# ONE-SHOT HIGH-FIDELITY IMITATION: TRAINING LARGE-SCALE DEEP NETS WITH RL

## ABSTRACT

Humans are experts at high-fidelity imitation – closely mimicking a demonstration, often in one attempt. Humans use this ability to quickly solve a task instance, and to bootstrap learning of new tasks. Achieving these abilities in autonomous agents is an open problem. In this paper, we introduce an off-policy RL algorithm (MetaMimic) to narrow this gap. MetaMimic can learn both (i) policies for high-fidelity one-shot imitation of diverse novel skills, and (ii) policies that enable the agent to solve tasks more efficiently than the demonstrators. MetaMimic relies on the principle of storing all experiences in a memory and replaying these to learn massive deep neural network policies by off-policy RL. This paper introduces, to the best of our knowledge, the largest existing neural networks for deep RL and shows that larger networks with normalization are needed to achieve one-shot high-fidelity imitation on a challenging manipulation task. The results also show that both types of policy can be learned from vision, in spite of the task rewards being sparse, and without access to demonstrator actions.

## 1 INTRODUCTION

One-shot imitation is a powerful way to show agents how to solve a task. For instance, one or a few demonstrations are typically enough to teach people how to solve a new manufacturing task. In this paper, we introduce an AI agent that when provided with a novel demonstration is able to (i) mimic the demonstration with high-fidelity, or (ii) forego high-fidelity imitation to solve the intended task more efficiently. Both types of imitation can be useful in different domains.

Motor control is a notoriously difficult problem, and we are often deceived by how simple a manipulation task might appear to be. Tying shoe-laces, a behaviour many of us learn by imitation, might appear to be simple. Yet, tying shoe-laces is something most 6 year olds struggle with, long after object recognition, walking, speech, often translation, and sometimes even reading comprehension. This long process of learning that eventually results in our ability to rapidly imitate many behaviours provides inspiration for the work in this paper.

We refer to *high-fidelity imitation* as the act of closely mimicking a demonstration trajectory, even when some actions may be accidental or irrelevant to the task. This is sometimes called over-imitation (McGuigan et al., 2011). It is known that humans over-imitate more than other primates (Horner & Whiten, 2005) and that this may be useful for rapidly acquiring new skills (Legare & Nielsen, 2015). For AI agents however, learning to closely imitate even one single demonstration from raw sensory input can be difficult. Many recent works focus on using expensive reinforcement learning (RL) methods to solve this problem (Sermanet et al., 2018; Liu et al., 2017; Peng et al., 2018; Aytar et al., 2018). In contrast, high-fidelity imitation in humans is often cheap: in one-shot we can closely mimic a demonstration. Inspired by this, we introduce a meta-learning approach (MetaMimic — Figure 1) to learn high-fidelity one-shot imitation policies by off-policy RL. These policies, when deployed, require a single demonstration as input in order to mimic the new skill being demonstrated.

AI agents could acquire a large and diverse set of skills by high-fidelity imitation with RL. However, representing many behaviours requires the adoption of a model with very high capacity, such as a very large deep neural network. Unfortunately, showing that RL methods can be used to train massive deep neural networks has been an open question because of the variance inherent to these methods. Indeed, traditional deep RL neural networks tend to be small, to the point that researchers have recently questioned their contribution (Rajeswaran et al., 2017b). In this paper, we show that it is possible to train massive

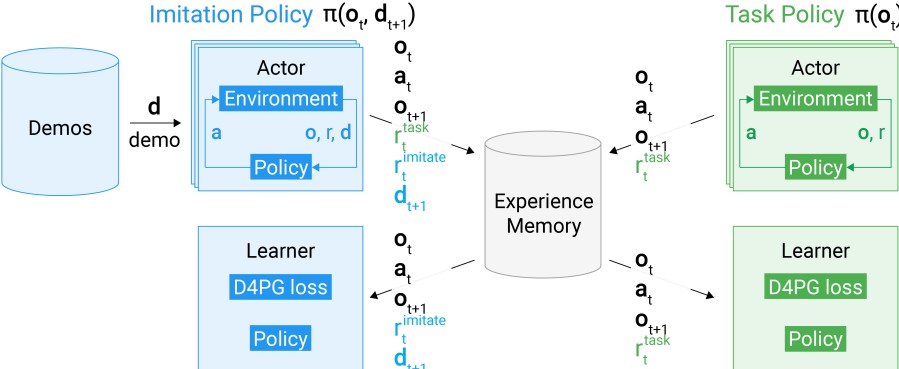

Figure 1: Starting from a dataset of demonstration videos, without expert actions, MetaMimic learns a *one-shot high-fidelity imitation policy* $\pi(\mathbf{o}_t, \mathbf{g}_t)$ with off-policy RL. This policy, represented with a massive deep neural network, enables the robot arm to mimic any demonstration in one-shot. In addition to producing an imitation policy that generalizes well, MetaMimic populates its replay memory with all its rich experiences, including not only the demonstration videos, but also its past observations, actions and rewards. By harnessing these augmented experiences, a *task policy* $\pi(\mathbf{o}_t)$ can be trained to solve difficult sparse-reward control tasks.

deep networks by off-policy RL to represent many behaviours. Moreover, we show that bigger networks generalize better. These results therefore provide important evidence that RL is indeed a scalable and viable framework for the design of AI agents. Specifically this paper makes the following contributions[1]:

- It introduces the MetaMimic algorithm and shows that it is capable of one-shot high-fidelity imitation from video in a complex manipulation domain.

- It shows that MetaMimic can harness video demonstrations and enrich them with actions and rewards so as to learn uncoditional policies capable of solving manipulation tasks more efficiently than teleoperating humans. By retaining and taking advantage of all its experiences, MetaMimic also substantially outperforms the state-of-the-art D4PG RL agent, when D4PG uses only the current task experiences.

- The experiments provide ablations showing that larger networks (to the best of our knowledge, the largest networks ever used in deep RL) lead to improved generalization in high-fidelity imitation. The ablations also highlight the important value of instance normalization.

- The experiments show that increasing the number of demonstrations during training leads to better generalization on one-shot high-fidelity imitation tasks.

## 2 METAMIMIC

MetaMimic is an algorithm to learn both one-shot high-fidelity imitation policies and unconditional task policies that outperform demonstrators. Component 1 takes as input a dataset of demonstrations, and produces (i) a set of rich experiences and (ii) a one-shot high-fidelity imitation policy. Component 2 takes as input a set of rich experiences and produces an unconditional task policy.

Component 1 uses a dataset of demonstrations to define a set of imitation tasks and using RL it trains a conditional policy to perform well across this set. Here each demonstration is a sequence of observations, without corresponding actions. This component can use any RL algorithm, and is applicable whenever the agent can be run in the environment and the environment's initial conditions can be precisely set. In practice we make use of D4PG (Barth-Maron et al., 2018), an efficient off-policy RL algorithm, for training the agent's policy from demonstration data. In Section 2.1 we give a detailed description of our approach for learning one-shot high-fidelity imitation policies. Here we also describe the neural network architectures used in our approach to imitation; our results will show that as the number of imitation tasks increases it becomes necessary to train large-scale neural network policies to generalize well. Furthermore, the process of training the imitation policies results in a memory of experiences which includes both actions and rewards. As shown in Section 2.2 we can replay these experiences to learn *unconditional policies* capable of solving new tasks and outperforming human demonstrators.

---

[1]Videos presenting our results are available at `https://vimeo.com/metamimic`.

**Algorithm 1** Imitation Actor

**Given:**
- an experience replay memory $\mathcal{M}$
- a dataset of demonstrations $\mathcal{D}$
- a reward function $r : \mathcal{O} \times \mathcal{O} \to \mathbb{R}$

Initialize memory $\mathcal{M}$
**for** $n_{\text{episodes}}$ **do**
    Sample demo $\mathbf{d}$ from $\mathcal{D}$
    Set initial state $\mathbf{o}_1$ to initial demo state $\mathbf{d}_1$
    **for** $t = 1$ **to** $T$ **do**
        Sample action $\mathbf{a}_t \leftarrow \pi(\mathbf{o}_t, \mathbf{d}_{t+1})$
        Execute $\mathbf{a}_t$ and observe $\mathbf{o}_{t+1}$ and $r_t^{\text{task}}$
        Calculate $r_t^{\text{imitate}} = r(\mathbf{o}_{t+1}, \mathbf{d}_{t+1})$
        Store $(\mathbf{o}_t, \mathbf{a}_t, \mathbf{o}_{t+1}, r_t^{\text{task}}, r_t^{\text{imitate}}, \mathbf{d}_{t+1})$ in $\mathcal{M}$
    **end for**
**end for**

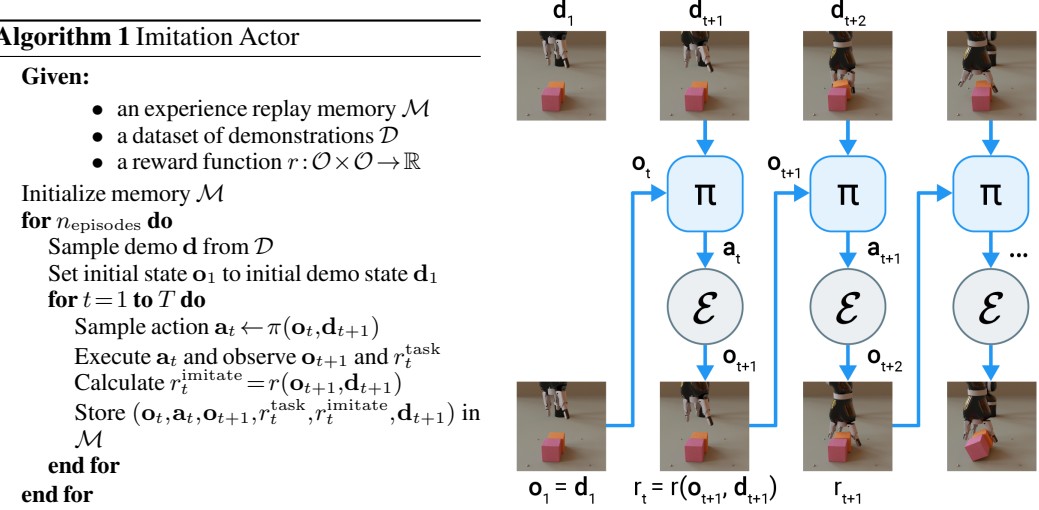

Figure 2: Imitation actor algorithm (left) and illustration of the environment and stochastic task (right). Algorithms for the imitation learner, the task actor and the task learner are provided in Appendix A.

## 2.1 LEARNING A POLICY FOR ONE-SHOT HIGH-FIDELITY IMITATION

We consider a single stochastic task and define a demonstration of this task as a sequence of observations $\mathbf{d} = \{\mathbf{d}_1, ..., \mathbf{d}_T\} \sim p_{\mathbf{d}}(\cdot)$. While there is only one task, there is ample diversity in the environment's initial conditions and in the demonstrations. Each observation $\mathbf{d}_t$ can be either an image or a combination of both images and proprioceptive information. We let $\pi_{\boldsymbol{\theta}}$ represent a deterministic parameterized imitation policy that produces actions $\mathbf{a}_t = \pi_{\boldsymbol{\theta}}(\mathbf{o}_t, \mathbf{d})$ by conditioning on the current observations and a given demonstration. We also let $\mathcal{E}$ denote the environment renderer, which accepts an arbitrary policy and produces a sequence of observations $\mathbf{o} = \{\mathbf{o}_1, ..., \mathbf{o}_T\} = \mathcal{E}(\pi)$. We can think of this last step as producing a rollout trajectory given the policy, as illustrated on the right side of Figure 2.

The goal of high-fidelity imitation is to estimate the parameters $\boldsymbol{\theta}$ of a policy that maximizes the expected imitation return, *e.g.* similarity between the observations $\mathbf{o}$ and sampled demonstrations $\mathbf{d} \sim p_{\mathbf{d}}$. That is,

$$\boldsymbol{\theta}^* = \operatorname*{argmax}_{\boldsymbol{\theta}} \mathbb{E}_{\mathbf{d} \sim p_{\mathbf{d}}} \left[ \sum_t \gamma^t \mathrm{sim}\Big( \mathbf{d}_t, \mathcal{E}\big(\pi_{\boldsymbol{\theta}}(\mathbf{o}_t, \mathbf{d})\big)_t \Big) \right], \tag{1}$$

where $\mathrm{sim}$ is a similarity measure (reward), which will be discussed in greater detail at the end of this section, and $\gamma$ is the discount factor. In general it is not possible to differentiate through the environment $\mathcal{E}$. We thus choose to optimize this objective with RL, while sampling trajectories by acting on the environment. Finally, we refer to this process as *one-shot* imitation because although we make use of multiple demonstrations at training time to learn the imitation policy, at test time we are able to follow a single novel demonstration $\mathbf{d}^\star$ using the learned conditional policy $\pi_{\boldsymbol{\theta}^*}(\mathbf{o}_t, \mathbf{d}^\star)$.

We adopt the recently proposed D4PG algorithm (Barth-Maron et al., 2018) as a subroutine for training the imitation policy. This is a distributed off-policy RL algorithm that interacts with the environment using independent actors (see Figure 2), each of which inserts trajectory data into a replay table. A learner process in parallel samples (possibly prioritized) experiences from this replay dataset and optimizes the policy in order to maximize the expected return. MetaMimic first builds on this earlier work by making a very specific choice of reward and policy.

At the beginning of every episode a single demonstration $\mathbf{d}$ is sampled, and the initial conditions of the environment are set to those of the demonstration, i.e. $\mathbf{o}_1 = \mathbf{d}_1$. The actor then interacts with the environment by producing actions $\mathbf{a}_t = \pi_{\boldsymbol{\theta}}(\mathbf{o}_t, \mathbf{d})$. While this policy representation is popular in the feature-based supervised one-shot imitation literature, in our case the observations and demonstrations are sequences of high-dimensional sensory inputs and hence this approach becomes computationally prohibitive. To overcome this challenge, we simplify the model to only consider local context $\mathbf{a}_t = \pi_{\boldsymbol{\theta}}(\mathbf{o}_t, \mathbf{d}_{t+1})$. In this formulation, the future demonstration state can be interpreted as a goal state and the approach may be thought of as goal-conditional imitation with a time-varying goal.

At every timestep, we compute the reward $r_t = r(\mathbf{o}_{t+1}, \mathbf{d})$. While in general this reward can depend on the entire trajectory—or on small subsequences such as $(\mathbf{d}_t, \mathbf{d}_{t+1})$—in practice we will restrict

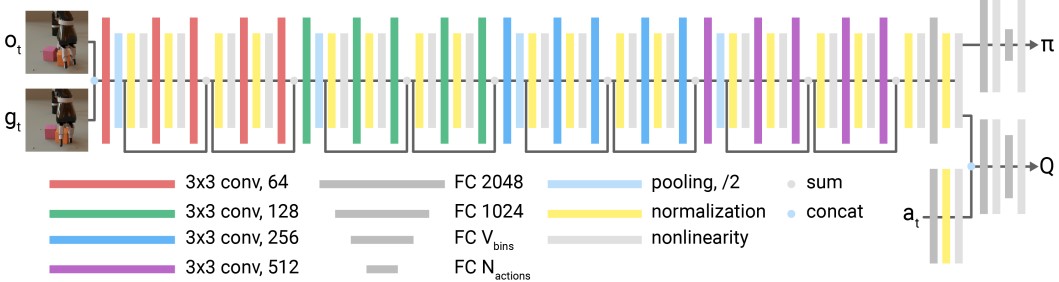

Figure 3: **Network architecture.** Since high-fidelity imitation is a fine-grained perception task, we found it necessary to use large convolutional neural networks. Our best network consists of a residual network (He et al., 2016) with twenty convolutional layers, instance normalization (Ulyanov et al.) between convolutional layers, layer normalization (Ba et al., 2016) between fully connected layers, and exponential linear units (Clevert et al., 2015). We use a similar network architecture for the imitation policy and task policy, however the task policy does not receive a goal $g_t$.

it to depend only on the goal state $\mathbf{d}_{t+1}$. Ideally the reward function can be learned during training (Ganin et al., 2018; Nair et al., 2018). However, in this work we experiment with a simple reward function based on the Euclidean distance[2] over observations:

$$r_t^{\text{imitate}} = r(\mathbf{o}_{t+1}, \mathbf{d}_{t+1}) = \beta_1 \exp\left(-\|\mathbf{o}_{t+1}^{\text{image}} - \mathbf{d}_{t+1}^{\text{image}}\|_2^2\right) + \beta_2 \exp\left(-\|\mathbf{o}_{t+1}^{\text{body}} - \mathbf{d}_{t+1}^{\text{body}}\|_2^2\right) \quad (2)$$

where $\mathbf{o}_t^{\text{image}}$ are the raw pixel observations and $\mathbf{o}_t^{\text{body}}$ are proprioceptive measurements (joint and end effector positions and velocities). Both components of the reward function have limitations: $\mathbf{o}_t^{\text{body}}$ has no information about objects in the environment, so it may fail to encourage the imitator to interact with the objects; where as $\mathbf{o}_t^{\text{image}}$ contains information about the body and objects, but is insufficient to uniquely describe either. In practice, we found a combination of both to work best.

Again we note that the next demonstration state $\mathbf{d}_{t+1}$ can be interpreted as a goal state. In this perspective the goals are set by the demonstration, and the agent is rewarded by the degree to which it reaches those goals. Because the imitation goals are explicitly given to the policy, the imitation policy is able to imitate many diverse demonstrations, and even generalize to unseen demonstrations as described in Section 2.2.

High-fidelity imitation is a fine-grained perception task and hence the choice of policy architecture is critical. In particular, the policy must be able to closely to mimic not only one but many possible ways of accomplishing the same stochastic task under different environment configurations. This representational demand motivates the introduction of high-capacity deep neural networks. We found the architecture, shown in Figure 3, with residual connections, 20 convolution layers with 512 channels for a total of 22 million parameters, and instance normalization to drastically improve performance, as shown in Figure 6 of the Experiments section. Following the recommendations of (Rajeswaran et al., 2017b), we compare the performance of this model with a smaller (15 convolution layers with 32 channels and 1.6 million) network proposed recently by Espeholt et al. (2018) and find size to matter. We note however that the IMPALA network of Espeholt et al. (2018) is large in comparison to previous networks used in important RL and control milestones, including AlphaGo (Silver et al., 2016), Atari DQN (Mnih et al., 2013), dexterous in-hand manipulation (OpenAI et al., 2018), QT-Opt for vision-based robotic manipulation (Kalashnikov et al., 2018), and DOTA among others.

## 2.2 LEARNING AN UNCONDITIONAL TASK POLICY

MetaMimic fills its replay memory with a rich set of experiences, which can also be used to learn a task more quickly. In order to train a task policy using RL we need to both explore, i.e. to find a sequence of actions that leads to high reward; and to learn, i.e. to harness reward signals to improve the generation of actions so as to generalize well. Unfortunately, stumbling on a rewarding sequence of actions for many control tasks is unlikely, especially when reward is only provided after a long sequence of actions.

A powerful way of using demonstrations for exploration is to inject the demonstration data directly into an off-policy experience-replay memory (Hester et al., 2017; Večerík et al., 2017; Nair et al., 2017b). However, these methods require access to privileged information about the demonstration – the sequences of actions and rewards – which is often not available. Our method takes a different approach.

---

[2]Ganin et al. (2018) show that $l^2$-distance is an optimal discriminator for conditional generation.

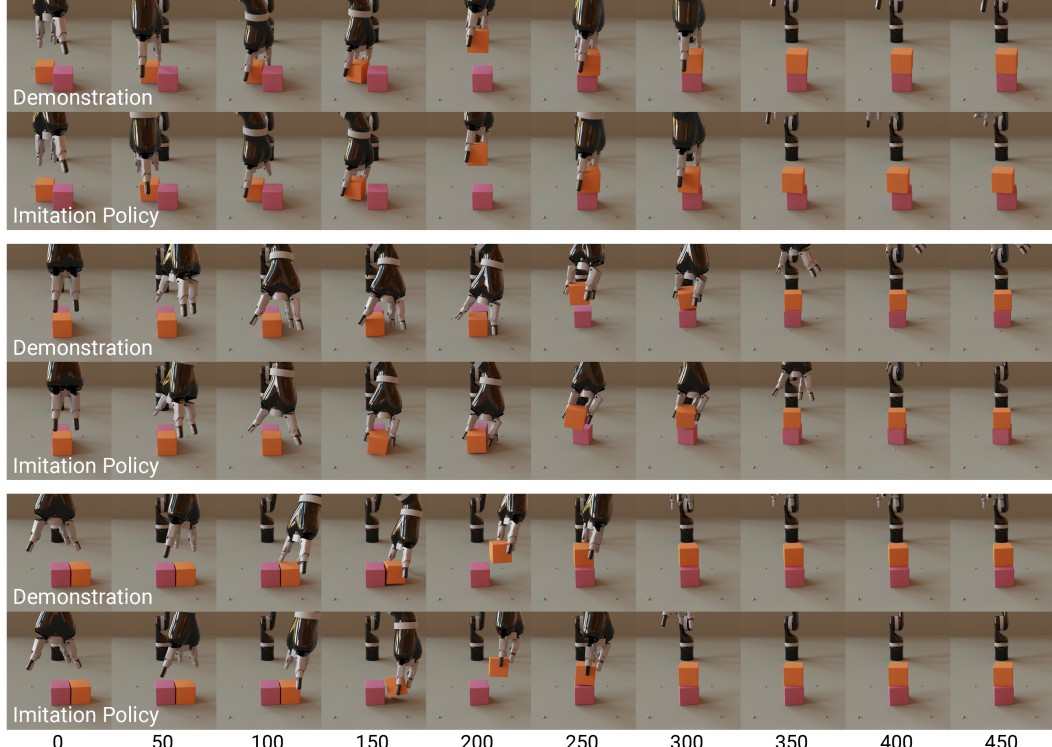

Figure 4: **One-shot high-fidelity imitation.** Given novel, diverse, test-set demonstration videos (three examples are shown above), the imitation policy is able to closely mimic the human demonstrator. In particular, it successfully maps image observations to arm velocities while managing the complex interaction forces among the arm, blocks and ground.

While our high-fidelity imitation policy attempts to imitate the demonstration from observations only, it generates its own observations, actions and rewards. These experiences are often rewarding enough to help with exploration. Therefore, instead of injecting demonstration trajectories, we place all experiences generated by our imitation policy in the experience-replay memory, as illustrated in Figure 1. The key design principle behind our approach is that RL agents should store all their experiences and take advantage of them for solving new problems.

More precisely, as the imitation policy interacts with the environment we also assume the existence of a *task reward* ($r_t^{\text{task}}$). Given these rewards we can introduce an additional, off-policy task learner which optimizes an unconditional task policy $\widetilde{\pi}_{\boldsymbol{\omega}}(\mathbf{o}_t)$. This policy can be learned from transitions $(\mathbf{o}_t, \mathbf{a}_t, \mathbf{o}_{t+1}, r_t^{\text{task}})$ generated asynchronously by the imitation actors following policy $\pi_{\boldsymbol{\theta}}(\mathbf{o}_t, \mathbf{d}_{t+1})$. This learning process is made possible by the fact that the task learner is simply optimizing the cumulative reward in an off-policy fashion. It should also be noted that this does not require privileged information about demonstrations because the sampled transitions are generated by the imitation actor's process of learning to imitate.

Due to the existence of demonstrations, the imitation trajectories are likely to lie in areas of high reward and as a result these samples can help circumvent the exploration problem. However, they are also likely to be very off-policy initially during learning. As a result, we augment these trajectories with samples generated asynchronously by additional task actors using the unconditional task policy $\widetilde{\pi}_{\boldsymbol{\omega}}(\mathbf{o}_t)$. The task learner then trains its policy by sampling from both imitation and task trajectories. For more algorithmic details see Appendix A. As the imitation policy improves, rewarding experiences are added to the replay memory and the task learner draws on these rewarding sequences to circumvent the exploration problem through off-policy learning. We will show this helps accelerate learning of the task policy, and that it works as well as methods that have direct access to expert actions and expert rewards.

## 3 EXPERIMENTS

In this section, we analyze the performance of our imitation and task policies. We chose to evaluate our methods in a particularly challenging environment: a robotic block stacking setup with both sparse

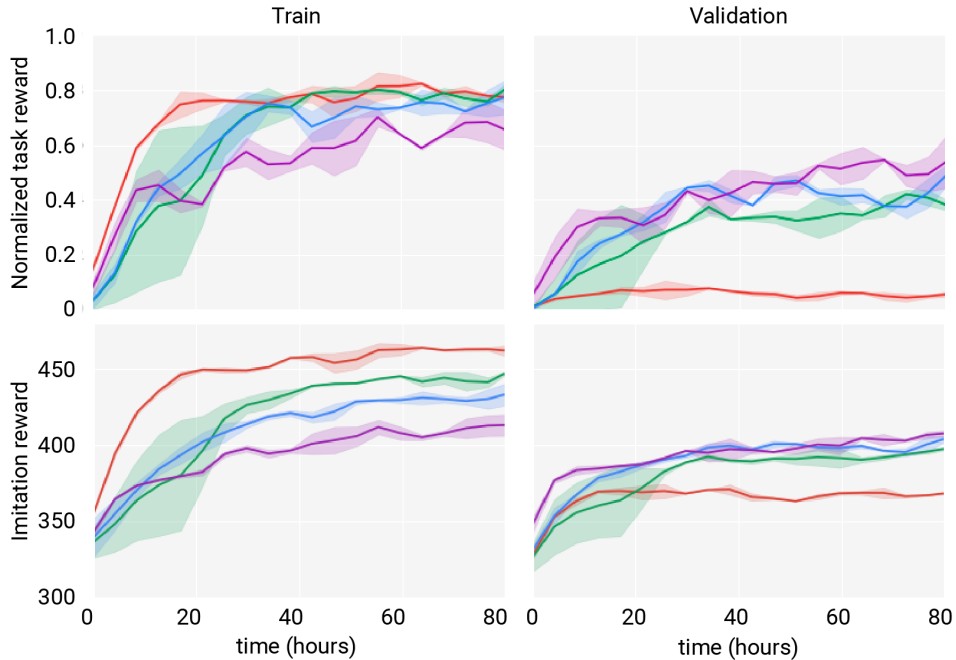

Figure 5: **More demonstrations improve generalization when using the imitation policy.** The goal of one-shot high-fidelity imitation is generalization to novel demonstrations. We analyze how generalization performance increases as we increase the number of demonstrations in the training set: 10 (—), 50 (—), 100 (—), 500 (—). With 10 demonstrations, the policy is able to closely mimic the training set, but generalizes poorly. With 500 demonstrations, the policy has more difficultly closely mimicking all trajectories, but has similar imitation reward on train and validation sets. The figure also shows that higher imitation reward when following the imitation policy (BOTTOM ROW) results in higher task reward (TOP ROW). Here we normalize the task reward by dividing it by the average of demonstration cumulative reward.

rewards and diverse initial conditions, learned from visual observations. In this space, our goal is to learn a policy performing *high-fidelity imitation* from human demonstration, while *generalizing* to new initial conditions.

### 3.1 ENVIRONMENT

Our environment consists of a Kinova Jaco arm with six arm joints and three actuated fingers, simulated in MuJoCo. In the *block stacking* task (Nair et al., 2017b), the robot interacts with two blocks on a tabletop. The task reward is a sparse piecewise constant function as described in (Zhu et al., 2018). The reward defines three stages (i) reaching, (ii) lifting, (iii) stacking. The reward only changes when the environment transitions from one stage to another. Our policy controls the simulated robot by setting the joint velocity commands, producing 9-dimensional continuous velocities in the range of $[-1,1]$ at 20Hz. The environment outputs the visual observation $\mathbf{o}_t^{\text{image}}$ as a $128 \times 128$ RGB image, as well as the proprioceptive features $\mathbf{o}_t^{\text{body}}$ consisting of positions and angular velocities of the arm and fingers joints.

In this environment, we collected demonstrations using a SpaceNavigator 3D motion controller, which allows human operators to control the robot arm with a position controller. We collected 500 episodes of demonstrations as imitation targets. Another 500 episodes were gathered for validation purposes by a different human demonstrator.

Note that the images shown in this paper have been rendered with the path-tracer Mitsuba for illustration purposes. Our agent does not however require such high-quality video input– the environment output generated by MuJoCo $\mathbf{o}_t^{\text{image}}$ which our agent observes is lower in resolution and quality.

### 3.2 HIGH-FIDELITY ONE-SHOT IMITATION

We use D4PG (see Appendix C.1) to train the imitation policy in a one-shot manner (sec. 2.1) on the block stacking task. The policy observes the visual input $\mathbf{o}_t^{\text{image}}$, as well a demonstration sequence $\mathbf{d}$ randomly sampled from the set of 500 expert episodes. In fig. 4 we show that our policy can closely mimic novel, diverse, test-set demonstration videos. Recall these test demonstrations are provided by

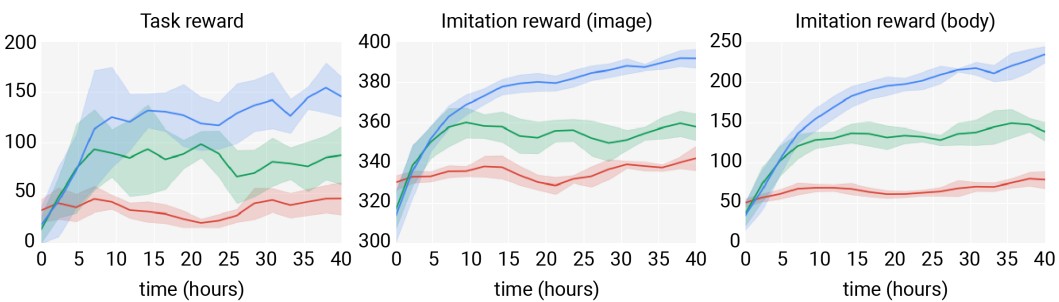

Figure 6: **Larger networks and normalization improve rewards in high-fidelity imitation.** We compare the ResNet model used by the IMPALA agent (15 conv layers, 32 channels) (—) with the much larger networks used by MetaMimic inspired by ResNet34 (20 conv layers, 512 channels) with (—) and without (—) instance normalization. We use three metrics for this comparison: task reward (LEFT), imitation reward when tracking pixels (MIDDLE) and imitation reward when tracking arm joint positions and velocities (RIGHT). We find large neural networks, and normalization significantly improve performance for high-fidelity imitation. To our knowledge, MetaMimic uses the largest neural network trained end-to-end using reinforcement learning.

a different expert and require generalization to a distinct stacking style. The test demonstrations are so different from the training ones that the average cumulative reward of the test demonstrations is lower than that of training by as much as 70. (The average episodic reward for the training demonstration set is 355 and that for the test set is 285.) On these novel demonstrations we are able to achieve 52% of the reward of the demonstration without any task reward, solely by doing high-fidelity imitation.

It is important to note that we achieve this while placing significantly less assumptions on the environment and demonstration than comparable methods. Unlike supervised methods (e.g. (Nair et al., 2017b)) we can imitate without actions at training time. And while proprioceptive features are used as part of computing the imitation reward, they are not observed by the policy, which means MetaMimic's imitation policy can mimic block stacking demonstrations purely from video at test time. And finally, as opposed to pure reinforcement-learning approaches ((Barth-Maron et al., 2018)) we do not train on a task reward.

**Generalization:** To analyze how well our learned policy generalizes to novel demonstrations, we run the policy conditioned on demonstrations from the validation set. As fig. 5 shows, validation rewards track the training curves fairly well. We also notice that policies trained on a small number of demonstrations achieve high imitation reward in training, but low reward on validation, while policies trained on a bigger set generalize much better. While we do not use a task reward in training, we use it to measure the performance on the stacking task. We see the same behavior as for the imitation reward: Policies trained on 50 or more demonstrations generalize very well. On the training set, performance varies from 67% of the average demonstration reward to 81% of the average demonstration reward.

**Network architecture:** Most reinforcement learning methods use comparatively small networks. However, high-fidelity imitation of this stochastic task requires the coordination of fine-grained perception and accurate motor control. These problems can strongly benefit from large architectures used for other difficult vision tasks. And due the fact that we are training with a dense reward, the training signal is rich enough to properly train even large models. In fig. 3 we demonstrate that indeed a large ResNet34-style network (He et al., 2016) clearly outperforms the network from IMPALA (Espeholt et al., 2018). Additionally, we show that using instance normalization (Ulyanov et al.) improves performance even more. We chose instance normalization instead of batch norm (Ioffe & Szegedy, 2015) to avoid distribution drift between training and running the policy (Ioffe, 2017). To our knowledge, this is the largest neural network trained end-to-end using reinforcement learning.

### 3.3    TASK POLICY

In the previous section we have shown that we are able to learn a high-fidelity imitation policy which mimics a novel demonstration in a single shot. It does however require a demonstration sequence as an input at test time. The task policy (see sec. 2.2), not conditioned on a demonstration sequence, is trained concurrently to the imitation policy, and learns from the imitation experiences along with its own experiences.

In fig. 7 we show a qualitative comparison of the task policy and a corresponding demonstration sequence. This is achieved by starting the task policy at the same initial state as the demonstration

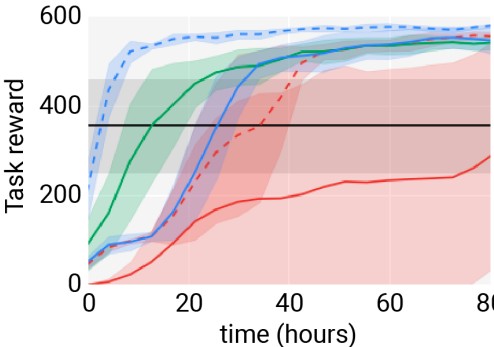

Figure 7: **Efficient task policy.** The task policy (which does not condition on a demonstration) is able to outperform the demonstration videos. We test this by initializing the environment to the same initial state as a demonstration from the training-set. The task policy is able to stack within 50 frames, while the demonstration stacks in 200 frames. The task policy has no incentive to behave like the demonstration, so it often lays its arm down after stacking.

Figure 8: **Comparing task policies.** We compare MetaMimic (—) to demonstrations (—), D4PG (—), and two methods that use demonstrations with access to additional information: D4PGfD (—), D4PG with a demonstration curriculum (- -), and MetaMimic with a curriculum (- -). D4PG is not able to reach the performance of the demonstrations. The methods with access to additional information are able to quickly outperform the demonstrators. MetaMimic matches their performance even without access to expert rewards or actions.

sequence. The task policy is not merely imitating the demonstration sequence, but has learned to perform the same task in much shorter time. For our task, the policy is able to outperform the demonstrations it learned from by 50% in terms of task reward (see fig. 8). Pure RL approaches are not able to reach this performance; the D4PG baseline scored significantly below the demonstration reward.

A really powerful technique in RL is to use demonstrations as a curriculum. To do that, one starts the episode during training such that the initial state is set to a random state along a random demonstration trajectory. This technique enables the agent to see the later and often rewarding part of a demonstration episode often. This approach has been shown to be very beneficial to RL as well as imitation learning (Resnick et al., 2018; Hosu & Rebedea, 2016; Popov et al., 2017). It, however, requires an environment which can be reset to any demonstration state, which is often only possible in simulation. We compare D4PG and our method both with and without a demonstration curriculum. As shown in previous results, using this curriculum for training significantly improves convergence for both methods. Our method without a demonstration curriculum performs as well as D4PG with the curriculum. When trained with the curriculum, our method significantly outperform all other methods; see fig. 8.

Last but not least, we compare our task policy against D4PGfD (Vecerík et al., 2017) (see sec. 4 for more details). D4PGfD differs from our approach in that it requires demonstration actions. While it takes time for our imitation policy to take off and help with training of the task policy, D4PGfD can help with exploration right away. It therefore is more efficient. Despite having no access to actions, however, our task policy catches up with D4PGfD quickly and reaches the same performance as the policy trained with D4PGfD. This speaks to the efficiency of our imitation policy. See fig. 8 for more details.

## 4 RELATED WORK

**General imitation learning:** Many prior works focus on using imitation learning to directly learn a task policy. There are two main approaches: Behavior cloning (BC) which attempts to learn a task policy by supervised learning (Pomerleau, 1989), and inverse RL (IRL) which attempts to learn a reward function from a set of demonstrations, and then uses RL to learn a task policy that maximizes that learned reward (Ng et al., 2000; Abbeel & Ng, 2004; Ziebart et al., 2008).

While BC has problems with accumulating errors over long sequences, it has been used successfully both on its own (Rahmatizadeh et al., 2017) and as an auxiliary loss in combination with RL (Rajeswaran et al., 2017a; Nair et al., 2017b). IRL methods do not necessarily require expert actions.

Generative Adversarial Imitation Learning (GAIL) (Ho & Ermon, 2016) is one example. GAIL constructs a reward function that measures the similarity between expert-generated observations and observations generated by the current policy. GAIL has been successfully applied in a number of different environments (Ho & Ermon, 2016; Li et al., 2017a; Merel et al., 2017; Zhu et al., 2018). While these methods work quite well, they focus on learning task policies, and not one-shot imitation.

**One-shot imitation learning:** Our approach is a form of one-shot imitation. A few recent works have explored one-shot task-based imitation learning (Finn et al., 2017; Duan et al., 2017; Wang et al., 2017), i.e. given a single demonstration, generalize to a new task instance with no additional environment interactions. These methods do not focus on high-fidelity imitation and therefore may not faithfully execute the same plan as the demonstrator at test time.

**Imitation by tracking:** Our method learns from demonstrations using a tracking reward (Atkeson & Schaal, 1997). This method has seen increased popularity in games (Aytar et al., 2018) and control (Sermanet et al., 2018; Liu et al., 2017; Peng et al., 2018). All these methods use tracking to imitate a single demonstration trajectory. Imitation by tracking has several advantages. For example it does not require access to expert actions at training time, can track long demonstrations, and is amenable to third person imitation (Sermanet et al., 2016). To our knowledge, MetaMimic is the first to train a single policy to closely track hundreds of demonstration trajectories, as well as generalize to novel demonstrations.

**Inverse dynamics models:** Our method is closely related to recent work on learned inverse dynamics models (Pathak et al., 2018; Nair et al., 2017a). These works train inverse dynamics models without expert demonstrations by self-supervision. However since these methods are based on random exploration they rely on high level control policies, structured exploration, and short horizon tasks. Torabi et al. (2018) also train an inverse dynamics model to learn an unconditional policy. Their method, however, uses supervised learning, and does not outperform BC.

**Multi-task off-policy reinforcement learning:** Our approach is related to recent work that learns a family of policies, with a shared pool of experiences (Sutton et al., 2011; Andrychowicz et al., 2017; Cabi et al., 2017; Riedmiller et al., 2018). This allows for sparse reward tasks to be solved faster, when paired with related dense reward tasks. Cabi et al. (2017) and Riedmiller et al. (2018) require the practitioner to design a family of tasks and reward functions related to the task of interest. In this paper, we circumvent the need of auxiliary task design via imitation.

**fD-style methods:** When demonstration actions are available, one can embed the expert demonstrations into the replay memory (Hester et al., 2017; Vecerík et al., 2017). Through off-policy learning, the demonstrations could lead to better exploration. This is similar to our approach as detailed in sec 2.2. We, however, eliminate the need for expert actions through high-fidelity imitation.

For a tabular comparison of the different imitation techniques, please refer to Table 2 in the Appendix.

# 5 CONCLUSIONS AND FUTURE WORK

In this paper, we introduced MetaMimic, a method to 1) train a high-fidelity one-shot imitation policy, and to 2) efficiently train a task policy. MetaMimic employs the largest neural network trained via RL, and works from vision, without the need of expert actions. The one-shot imitation policy can generalize to unseen trajectories and can mimic them closely. Bootstrapping on imitation experiences, the task policy can quickly outperform the demonstrator, and is competitive with methods that receive privileged information.

The framework presented in this paper can be extended in a number of ways. First, it would be exciting to combine this work with existing methods for learning third-person imitation rewards (Sermanet et al., 2016; 2018; Aytar et al., 2018). This would bring us a step closer to how humans imitate: By watching other agents act in the environment. Second, it would be exciting to extend MetaMimic to imitate demonstrations of a variety of tasks. This may allow it to generalize to demonstrations of unseen tasks.

To improve the ease of application of MetaMimic to robotic tasks, it would be desirable to address the question of how to relax the initialization constraints for high-fidelity imitation; specifically not having to set the initial agent observation to be close to the initial demonstration observation.

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

## A   ADDITIONAL ALGORITHM DETAILS

---

**Algorithm 2** Imitation Learner

---

**Given:**
- an off-policy RL algorithm $\mathbb{A}$
- a replay buffer $\mathcal{M}$

Initialize $\mathbb{A}$
**for** $n_{updates}$ **do**
    Sample transitions $(\mathbf{o}_t, \mathbf{a}_t, \mathbf{o}_{t+1}, r_t^{\text{imitate}}, \mathbf{d}_{t+1})$ from $\mathcal{M}$ to make a minibatch $B$.
    Perform on update step using $\mathbb{A}$ and $B$.
**end for**

---

**Algorithm 3** Task Actor

---

**Given:**
- an experience replay memory $\mathcal{M}$

Initialize memory $\mathcal{M}$
**for** $n_{episodes}$ **do**
    **for** $t=1$ **to** $T$ **do**
        Sample action from task policy: $\mathbf{a}_t \leftarrow \pi(\mathbf{o}_t)$
        Execute action $\mathbf{a}_t$ and observe new state $\mathbf{o}_{t+1}$, and reward $r_t^{\text{task}}$
        Store transition $(\mathbf{o}_t, \mathbf{a}_t, \mathbf{o}_{t+1}, r_t^{\text{task}})$ in memory $\mathcal{M}$
    **end for**
**end for**

---

**Algorithm 4** Task Learner

---

**Given:**
- an off-policy RL algorithm $\mathbb{A}$
- a replay buffer $\mathcal{M}$

Initialize $\mathbb{A}$
**for** $n_{updates}$ **do**
    Sample transitions $(\mathbf{o}_t, \mathbf{a}_t, \mathbf{o}_{t+1}, r_t^{\text{task}})$ from $\mathcal{M}$ to make a minibatch $B$.
    Perform on update step using $\mathbb{A}$ and $B$.
**end for**

---

## B    HYPERPARAMETERS

Table 1 lists the hyperparameters used for the experiments.

Table 1: Hyper-parameters used for all experiments.

| Parameters | Values | Comments |
|---|---|---|
| Image Width | 128 | Did not tune |
| Image Height | 128 | Did not tune |
| **D4PG Parameters** | | |
| $V_{min}$ | 0 | Important |
| $V_{max}$ | 100 | Important |
| $V_{bins}$ | 101 | Did not tune |
| N step | 5 | Did not tune |
| Actor learning rate | loguniform([5e-5, 2e-4]) | – |
| Critic learning rate | loguniform([5e-5, 2e-4]) | – |
| Optimizer | Adam Kingma & Ba (2014) | Did not tune |
| Batch size | 64 | Did not tune |
| Target update period | 100 | Did not tune |
| Discount factor ($\gamma$) | 0.99 | Did not tune |
| Replay capacity | 1e6 | Did not tune |
| **Imitation Parameters** | | |
| Early termination cutoff | 0.5 | Important |
| Pixel reward coefficient ($\beta_1$) | 15 | – |
| Joint reward coefficient ($\beta_2$) | 2 | – |
| Imitation actors | 256 | – |
| Demo as a curriculum | – | See experiments |
| Sampling | Prioritized | Important |
| Removal | Prioritized | Important |
| **Task Parameters** | | |
| Task actors | 256 | – |
| Sampling | Uniform | Did not tune |
| Removal | First-in-first-out | Did not tune |

## C    TRAINING DETAILS

### C.1    D4PG

We use D4PG (Barth-Maron et al., 2018) as our main training algorithm. Briefly, D4PG is a distributed off-policy reinforcement learning algorithm for continuous control problems. In a nutshell, D4PG uses Q-learning for policy evaluation and Deterministic Policy Gradients (DPG) (Silver et al., 2014) for policy optimization. An important characteristic of D4PG is that it maintains a replay memory $\mathcal{M}$ (possibility prioritized (Horgan et al., 2018)) that stores SARS tuples which allows for off-policy learning. D4PG also adopts target networks for increased training stability. In addition to these principles, D4PG utilized distributed training, distributional value functions, and multi-step returns to further increase efficiency and stability. In this section, we explain the different ingredients of D4PG.

D4PG maintains an online value network $Q(\mathbf{o},\mathbf{a}|\boldsymbol{\theta})$ and an online policy network $\pi(\mathbf{o}|\boldsymbol{\phi})$. The target networks are of the same structures as the value and policy network, but are parameterized by different parameters $\boldsymbol{\theta}'$ and $\boldsymbol{\phi}'$ which are periodically updated to the current parameters of the online networks.

Given the $Q$ function, we can update the policy using DPG:

$$\mathcal{J}(\boldsymbol{\phi}) = \mathbb{E}_{\mathbf{o}_t \sim \mathcal{M}}\left[\nabla_{\boldsymbol{\phi}} Q(\mathbf{o}_t, \pi(\mathbf{o}_t|\boldsymbol{\phi}))\right].$$

Instead of using a scalar $Q$ function, D4PG adopts a distributional value function such that $Q(\mathbf{o}_t,\mathbf{a}|\boldsymbol{\theta}) = \mathbb{E}\left[Z(\mathbf{o}_t,\mathbf{a}|\boldsymbol{\theta})\right]$ where $Z$ is a random variable such that $Z = z_i$ w.p. $p_i \propto \exp(\omega(\mathbf{o}_t,\mathbf{a}|\boldsymbol{\theta}))$.

The $z_i$'s take on $V_{bins}$ discrete values that ranges uniformly between $V_{min}$ and $V_{max}$ such that $z_i = V_{min} + i \frac{V_{max} - V_{min}}{V_{bins}}$ for $i \in \{0, \cdots, V_{bins} - 1\}$.

To construct a bootstrap target, D4PG uses N-step returns. Given a sampled tuple from the replay memory: $\mathbf{o}_t, \mathbf{a}_t, \{r_t, r_{t+1}, \cdots, r_{t+N-1}\}, \mathbf{o}_{t+N}$, we construct a new random variable $Z'$ such that $Z' = z_i + \sum_{n=0}^{N-1} \gamma^n r_{t+n}$ w.p. $p_i \propto \exp(\omega(\mathbf{o}_t, \mathbf{a} | \boldsymbol{\theta}'))$. Notice, $Z'$ no longer has the same support. We therefore adopt the same projection $\Phi$ employed by Bellemare et al. (2017). The training loss for the value fuction

$$\mathcal{L}(\boldsymbol{\theta}) = \mathbb{E}_{\mathbf{o}_t, \mathbf{a}_t, \{r_t, \cdots, r_{t+N-1}\}, \mathbf{o}_{t+N} \sim \mathcal{M}} \big[ H(\Phi(Z'), Z(\mathbf{o}_t, \mathbf{a}_t | \boldsymbol{\theta})) \big],$$

where $H$ is the cross entropy.

Last but not least, D4PG is also distributed following Horgan et al. (2018). Since all learning processes only rely on the replay memory, we can easily decouple the 'actors' from the 'learners'. D4PG therefore uses a large number of independent actor processes which act in the environment and write data to a central replay memory process. The learners could then draw samples from the replay memory for learning. The learner also serves as a parameter server to the actors which periodically update their policy parameters from the learner. For more details see Algorithms 1-4.

## C.2 FURTHER DETAILS

When using a demonstration curriculum, we randomly sample the initial state from the first 300 steps of a random demonstration.

For the Jaco arm experiments, we consider the vectors between the hand and the target block for the environment and the demonstration and compute the L2 distance between these. The episode terminates if the distance exceeds a threshold (0.01).

| Methods | Use Demo Actions | Rewards | One-Shot Imitation | High-Fidelity Imitation | Unconditional Imitation | BC vs. RL |
|---|---|---|---|---|---|---|
| DQfD / DDPGfD (Hester et al., 2017) (Vecerík et al., 2017) | Yes | Yes | No | No | Yes | RL |
| Nair et al. (2017b) | Yes | Yes | No | No | Yes | RL |
| Rajeswaran et al. (2017a) | Yes | Yes | No | No | Yes | RL |
| GAIL (Ho & Ermon, 2016) | No | No | No | No | Yes | RL |
| MaxEnt IRL (Ziebart et al., 2008) (Finn et al., 2016) | No | No | No | No | Yes | RL |
| Wang et al. (2017) | No | No | Yes | No | No | RL |
| Infogail Li et al. (2017b) | No | No | Yes | No | No | RL |
| DeepMimic (Peng et al., 2018) | No | No | No | No | N/A | RL |
| Unsupervised Perceptual Rewards (Sermanet et al., 2016) | No | No | No | No | Yes | RL |
| Aytar et al. (2018) | No | No | No | No | N/A | RL |
| One-shot imitation (Duan et al., 2017) (Finn et al., 2017) | Yes | No | Yes | No | No | BC |
| Zero-shot imitation Pathak et al. (2018) | No | No | Yes | Yes | No | BC |
| Liu et al. (2017) | No | No | Yes | Yes | No | BC |
| **Ours** (Imitation) | No | No | Yes | Yes | No | RL |
| **Ours** (Task) | No | Yes | No | No | Yes | RL |
| **Ours** (MetaMimic) | No | Yes | Yes | Yes | Yes | RL |

Table 2: A table comparing some imitation approaches. The columns describe different capabilities. The red color indicates a less desirable property and the green color differentiates between behavior cloning versus RL based methods.

