# OpenReview forum: "One-Shot High-Fidelity Imitation: Training Large-Scale Deep Nets with RL"
_ICLR.cc/2019/Conference_

### Official Review · AnonReviewer3 · 2018-11-07
**Well presented, but not suitable for ICLR**

**Rating:** 5
**Confidence:** 3

**Review:**


Summary:
This paper proposes MetaMimic, an algorithm that does the following:
(i) Learn to imitate with high-fidelity with one-shot. The setting is that we have access to several demonstrations (only states, no actions) of the same task. During training, we have pixel observations plus proprioceptive measurements). At test time, the learned policy can imitate a single new demonstration (consisting of only pixel observations) of the same task.
(ii) When given access to rewards, the policy can exceed the human demonstrator by augmenting its experience replay buffer with the experience gained while learning (i). Therefore, even in a setting with sparse rewards and no access to expert actions (only states), the policy can learn to solve the task.

Overall Evaluation:
This is a good paper. In my opinion however, it does not pass the bar for ICLR.

Pros:
- The paper is well written. The contributions are clearly listed, the methods section is easy to follow and the authors explain the choices they make. The illustrations are clear and intuitive.
- The overview of hyperparameter choice and tuning / importance factor in the Appendix is useful.
- Interesting pipeline of learning policies that can use demonstrations without actions.
- The results on the simulated robot arm (block stacking task with two blocks) are good.

Cons:
- The abstracts oversells the contribution a bit when saying that MetaMimic can learn "policies for high-fidelity one-shot imitation of diverse novel skills". The setting that's considered in the paper is that of a single task, but different demonstrations (different humans from different starting points). This seems restrictive, and could have been motivated better.
- Experimental results are shown only for one task; block stacking with a robot arm in simulation.
- Might not be a good topical fit for ICLR, but more suited for a conference like CoRL or a workshop. The paper is very specific to imitation learning for a manipulation / control tasks, where we can (1) reset the environment to the exact starting position of the demonstrations, (2) the eucledian distance between states in the demonstration and visited by the policy is meaningful (3) we have access to both pixel observations and proprioceptive measurements. The proposed method is an elegant way to solve this, but it's unclear how well it would perform on different types of control problems, or when we want to transfer policies between different (but related) tasks.

Questions:
- Where does the "task stochasticity" come from? Only from the starting state, and from having different demonstrations? Or is the transition function also stochastic?
- The learned policy is able to do one-shot imitation, i.e., given a new demonstration (of the same task) the policy can follow this demonstration. Do I understand correct that this mean that there is *no* additional learning required at test time?
- It is not immediately clear to me why the setting of a single task but new demonstrations is interesting. Could the authors comment on this? One setting I could imagine is that the policy is trained in simulation, but then executed in the real-world, given a new demonstration. (If that's the main motivation though, then the experiments might have to support that this is possible - if no real-world robot is available, maybe the same simulator with a slightly different camera angle / light conditons or so.)
- The x-axis in the figures says "time (hours)" - is that computation time or simulated time?

Other Comments:
- In 3.2, I would be interested in seeing the following baseline comparison: Learn the test task from scratch using the one available demonstration, with the RL procedure (Equation 2, but possibly without the second term to make it fair). In Figure 5, we can see that the performance on the training tasks is much better when training on only 10 tasks, compared to 500. Then why not overfit to a single task, if that's what we're interested in?
- An interesting baseline for 3.3 might be an RL algorithm with shaped rewards: using an additional reward term that is the eucledian distance to the *closest* datapoint from the demonstration. Compared to the baselines shown in the results section, this would be a fairer comparison because (1) unlike D4PG we also have access to information from the demonstrations and (2) no additional information is needed like the action information in D4PGfD and (3) we don't have the need for a curriculum.

Nitpick (no influence on score):
[1. Introduction]
- I find the first sentence, "One-shot imitation is a powerful way to show agents how to solve a task" a bit confusing. I'd say one-shot imitation is a method, not a way to show how to solve a task. Maybe an introductory sentence like "Expert demonstrations are a powerful way to show agents how to solve a task." works better?
- Second sentence, the chosen example is "manufacturing" tasks - do you mean manipulation? When reading this, I had to think of car manufacturing - a task I could certainly not imitate with just a few demonstrations.
- Add note that with "unconditional policy" you mean not conditioned on a demonstration.
[2. MetaMimic]
- [2.1] Third paragraph: write "Figure 2, Algorithm 1" or split the algorithm and figure up so you can refer to them separately.
- [2.1] Last paragraph, second line: remove second "to"

---

> ### Author Response · Authors · 2018-11-24
> **Authors' response to reviewer 3 part 1**
>
> Thank you very much for your feedback. We address your comments below.
>
> > The abstracts oversells the contribution a bit when saying that MetaMimic can learn "policies for high-fidelity one-shot imitation of diverse novel skills". The setting that's considered in the paper is that of a single task, but different demonstrations (different humans from different starting points). This seems restrictive, and could have been motivated better.
>
> We fully agree. We should have been more specific, e.g.  "policies for high-fidelity one-shot imitation of diverse novel motions in block stacking". We plan to be much more specific in the writeup explaining the sources of variation in the task.
>
> > Experimental results are shown only for one task; block stacking with a robot arm in simulation.
>
> This is correct. However there is significant variation in the motions, and since we are interested in high-fidelity imitation (and not just imitation for the purposes of solving the task), we believe this is a significant source of variation. This is in line with our experiment showing that one needs a neural net with very large capacity (the largest ever trained with RL) to generalize to novel test demonstrations in high-fidelity imitation.
>
> > Might not be a good topical fit for ICLR, but more suited for a conference like CoRL or a workshop. The paper is very specific to imitation learning for a manipulation / control tasks, where we can (1) reset the environment to the exact starting position of the demonstrations, (2) the eucledian distance between states in the demonstration and visited by the policy is meaningful (3) we have access to both pixel observations and proprioceptive measurements. The proposed method is an elegant way to solve this, but it's unclear how well it would perform on different types of control problems, or when we want to transfer policies between different (but related) tasks.
>
> We feel there are important questions of representation here that make the work interesting for ICLR. For instance, prior to this work we didn’t know we could train such massive neural nets with RL, and that increasing the size of these particular models is needed for generalization. We fully agree that (1) is an important limitation of the present approach. We were more interested in motion variation (the style in which any user solves a specific task) that on say object variation. Our focus is on high-fidelity imitation - if the focus is on imitation, then the techniques of pointed out in our reply to reviewer 4 are better choices (eg the works of Silvio Savarese, Chelsea Finn and colleagues.
>
> > Where does the "task stochasticity" come from? Only from the starting state, and from having different demonstrations? Or is the transition function also stochastic?
>
> Correct, the task stochasticity only comes from the starting state, and from having different demonstrations. We called it stochastic to distinguish it from environments like atari which are completely deterministic, i.e. without different starting conditions or different goals to achieve.
>
> > The learned policy is able to do one-shot imitation, i.e., given a new demonstration (of the same task) the policy can follow this demonstration. Do I understand correct that this mean that there is *no* additional learning required at test time?
>
> Yes, that is correct. At test time, the imitation policy is able to follow (never-seen-before) trajectories with no additional learning very closely. This is what we believe is a very cool result, especially because other groups have struggled to achieve this.
>
> > It is not immediately clear to me why the setting of a single task but new demonstrations is interesting. Could the authors comment on this? One setting I could imagine is that the policy is trained in simulation, but then executed in the real-world, given a new demonstration. (If that's the main motivation though, then the experiments might have to support that this is possible - if no real-world robot is available, maybe the same simulator with a slightly different camera angle / light conditons or so.)
>
> We would argue it is interesting because it is a skill that humans have, that is nontrivial for agents. Humans can observe a demonstration and imitate it very closely using just observations. In a factory, a manager might demonstrate to a new worker what to do with a set of objects, and then the new worker repeats the task with the same objects. Of course, humans can do this in a much more general way: e.g. from a third person perspective, and with abstract notions of perceptual similarity. Admittedly, we are all far from solving the full problem.
>
> Still, we think this is an interesting and useful step in that direction. One that demonstrates learning a complex mapping from perception to motor control through experience with an environment.
>
> We agree that the sim-to-real version of the problem is quite interesting.

---

> > ### Comment · AnonReviewer3 · 2018-11-27
> > **reply**
> >
> > Thank you for your reply and clarifications on the points I raised. Overall I still think this is an interesting approach, and an elegant way of dealing with the fact that actions are not available from the demonstrations. I stand by my evaluation, and think that the current version of the paper does not pass the bar for the main track of ICLR.
> >
> > It's interesting to hear about the two experiments that you tried and that did not work. I would suggest putting those in the Appendix, together with your hypothesis on why this does not work. These insights can be very valuable for other researchers.
> >
> > An updated version of the paper should, in my view, have a clearer motivation for why we are interested in high fidelity imitation of the same task but different demonstrators.
> >
> > I also think future work in this direction, but with different tasks instead of different demonstrators, sounds promising.

---

> ### Author Response · Authors · 2018-11-24
> **Authors' response to reviewer 3 part 2**
>
> > The x-axis in the figures says "time (hours)" - is that computation time or simulated time?
>
> The x-axis here refers to computation time.
>
> > In 3.2, I would be interested in seeing the following baseline comparison: Learn the test task from scratch using the one available demonstration, with the RL procedure (Equation 2, but possibly without the second term to make it fair). In Figure 5, we can see that the performance on the training tasks is much better when training on only 10 tasks, compared to 500. Then why not overfit to a single task, if that's what we're interested in?
>
> We have run a similar experiment before, training both policies with varying number of demonstrations. With 10 demonstrations, the task policy still learns quickly, but achieves lower reward, and qualitatively is more cumbersome, repeatedly attempting to stack one block atop the other until it finds a stable position. Often it drops the block and has to pick it back up. As the number of demonstrations increases, the max reward reached increases, and the learned task policy stacks in one smooth motion. We will rerun this experiment and have plots for the camera ready.
>
> > An interesting baseline for 3.3 might be an RL algorithm with shaped rewards: using an additional reward term that is the eucledian distance to the *closest* datapoint from the demonstration. Compared to the baselines shown in the results section, this would be a fairer comparison because (1) unlike D4PG we also have access to information from the demonstrations and (2) no additional information is needed like the action information in D4PGfD and (3) we don't have the need for a curriculum.
>
> We actually tried a related method without much success. Every episode we sampled a demonstration, and trained an unconditional policy that was rewarded for reaching the next step of the demonstration (not knowing which demonstration was provided). This method did not take off at all, because the policy did not know what goal it was trying to reach.
>
> The suggested variation may work better because the policy will be rewarded as long as it reaches any goal. We are concerned that a trivial solution would be to take no action and stay at the same observation. Alternatively it could also oscillate back and forth between two observations. Nevertheless it is an interesting experiment and we hope to have this in time for the camera-ready. Thanks for suggesting it.
>
> > I find the first sentence, "One-shot imitation is a powerful way to show agents how to solve a task" a bit confusing. I'd say one-shot imitation is a method, not a way to show how to solve a task. Maybe an introductory sentence like "Expert demonstrations are a powerful way to show agents how to solve a task." works better?
>
> Totally agree. We’ll update the text as you suggest.
>
> > Second sentence, the chosen example is "manufacturing" tasks - do you mean manipulation? When reading this, I had to think of car manufacturing - a task I could certainly not imitate with just a few demonstrations.
>
> Indeed. We will clarify.
>
> > Add note that with "unconditional policy" you mean not conditioned on a demonstration.
> [2. MetaMimic]
> - [2.1] Third paragraph: write "Figure 2, Algorithm 1" or split the algorithm and figure up so you can refer to them separately.
> - [2.1] Last paragraph, second line: remove second "to"
>
> We will make the changes suggested in this section. Thanks for the helpful suggestions, and for having devoted your time to clearly understand our paper and provide constructive feedback.

---

### Official Review · AnonReviewer1 · 2018-11-11
**learning from video demonstration; exposition is confusing / misleading.**

**Rating:** 5
**Confidence:** 3

**Review:**

This paper presents an RL method for learning from video demonstration without access to expert actions. The agent first learn to imitate the expert demonstration (observed image sequence and proprioceptive information) by producing a sequence of actions that will lead to the similar observations (require a renderer that takes actions and outputs images). The imitation loss is a similarity metric. Next, the agent explores the environment with both the imitation policy and task policy being learned; an off-policy RL algorithm D4PG is used for policy learning. Experiments are conducted on a simulated robot block stacking task.

The paper is really clearly written, but presenting the approach as "high-fidelity", "one-shot" learning is a bit confusing. First, it's not clear what's the motivation for high-fidelity. To me this is an artifact due to having to imitate the visual observation instead of the actions, which is a legitimate constraint, but not the original goal. Second, the one-shot learning setting consists of training on a set of stochastic demonstrations and testing on another set collected from a different person; both for the same task. Usually one-shot learning tests on slightly different tasks or environments, whereas here the goal is to generalize to novel demonstrations. It's not clear why do we care imitation per se in addition to the task reward.

What I find interesting is the proposed approach for learning for video demonstration without action labels. Currently this requires an executor to render the actions to images, what if we don't have such an executor or only have a noisy / approximate renderer? In the real world it's probably hard to find a good renderer, it would be interesting to see how this constraint can be relaxed.

Questions:
- While the authors have shown the average rewards of the two sets are different, I wonder what's the variance of each person's demonstration.
- In Fig 5, on the validation set, in terms of imitation loss there aren't that much difference between the policies, but in terms of task reward, the 'red' policy goes to zero while others policies' rewards are still similar. Any intuition for why seemingly okay imitation doesn't translate to task reward?

Overall, I enjoyed reading the paper and the experiments are comprehensive. The current presentation angle seems a bit off though.

---

> ### Author Response · Authors · 2018-11-24
> **Authors' response to reviewer 1**
>
> Thank you for the valuable feedback. We will address some of your comments and questions below.
>
> > The paper is really clearly written, but presenting the approach as "high-fidelity", "one-shot" learning is a bit confusing. First, it's not clear what's the motivation for high-fidelity. To me this is an artifact due to having to imitate the visual observation instead of the actions, which is a legitimate constraint, but not the original goal. Second, the one-shot learning setting consists of training on a set of stochastic demonstrations and testing on another set collected from a different person; both for the same task. Usually one-shot learning tests on slightly different tasks or environments, whereas here the goal is to generalize to novel demonstrations. It's not clear why do we care imitation per se in addition to the task reward.
>
> Given the review scores, we can only agree with you that the paper is somewhat confusing and we have failed to motivate high-fidelity imitation properly. We think that what makes the paper confusing is that as it stands it tells two stories (Hi-Fi imitation and task policies). These two stories are fundamentally linked, however we admit the presentation did not make these links clear. We will try to address this, but we very much look forward to any advice on how to change the presentation to make it more understandable.
>
> We used to the terminology One-Shot High-Fidelity Imitation to clarify both how our method works, and how it differs from existing methods in the space. First, high-fidelity is about mimicking the trajectory precisely. In precision engineering or surgery, where we don’t want the actuator doing anything other than what was demonstrated, this seems like a valuable idea. Existing few-shot imitation works focus on solving tasks, but not on following the trajectory precisely (see for example https://bair.berkeley.edu/blog/2018/06/28/daml/ ). In relation to this work, Yu and Finn point out: “While our work enables one-shot learning for manipulating new objects from one video of a human, our current experiments do not yet demonstrate the ability to learn entirely new motions in one shot”. The latter is what is demonstrated in our generalization experiments. That is, given a new demonstration motion, our policy is able to follow it closely as shown in Figure 4.
>
> Second, we use the phrase "one-shot" to distinguish our method from other tracking based methods, which can require many thousands of environment interactions to learn to track a single trajectory. In contrast, our method requires no additional environment interactions to track a novel trajectory. It achieves this in the same way one-shot methods do, by learning a policy that works well across a large dataset of "tasks" where each "task" is a demonstration.
>
> > What I find interesting is the proposed approach for learning for video demonstration without action labels. Currently this requires an executor to render the actions to images, what if we don't have such an executor or only have a noisy / approximate renderer? In the real world it's probably hard to find a good renderer, it would be interesting to see how this constraint can be relaxed.
>
> We agree that relaxing the constraint of an exact environmental renderer is an interesting research direction. This would be helpful for our method, as well as many other RL based methods. But we think it is beyond the scope of this paper.
>
> > While the authors have shown the average rewards of the two sets are different, I wonder what's the variance of each person's demonstration.
>
> There is notable variance between the two demonstrators, and between each demonstration. We will provide some additional examples in the appendix.
>
> > In Fig 5, on the validation set, in terms of imitation loss there aren't that much difference between the policies, but in terms of task reward, the 'red' policy goes to zero while others policies' rewards are still similar. Any intuition for why seemingly okay imitation doesn't translate to task reward?
>
> Yes, we have noticed two types of behavior that have reasonably high imitation rewards, but do not successfully complete the task: (i) the policy closely imitates the arm but ignores the block entirely, (ii) the policy successfully imitates both the arm and block position in the beginning of the trajectory, but fails to place the block on a stable position during the stack. As the imitation reward increases we see these behaviors less.
>
> > Overall, I enjoyed reading the paper and the experiments are comprehensive. The current presentation angle seems a bit off though.
>
> Thanks! We are really glad you enjoyed the paper, and the experiments, and hope we can align the presentation a bit better.

---

### Official Review · AnonReviewer2 · 2018-11-12
**Insufficient evidence/experimental validation for the main claims of the paper**

**Rating:** 4
**Confidence:** 4

**Review:**

Summary

This work porposes a approach for one-shot imitation with high accuracy, called "high fidelity imitation learning" by the authors. Furthermore, the work addresses the common problem of exploration in imitation learning, which would help to rescue from off-policy states.

Review

In my opinion, the main claims of this paper are not validated sufficiently in the experiments. I would expect the experiments to be designed specifically to support the claims made, but little evidence is provided:

- The authors claim that the method allows one-shot generalization to an unknown trajectory. To test this hypothesis the authors only provide experiments of generalization towards trajectories of a different demonstrator on the same task of stacking cubes. I would expect experiments with truly different trajectories on a different task than stacking cubes to test the hypothesis of one-shot imitation.
Until then I see no evidence for a "one-shot" imitation capability of the proposed method.

- That storing the trajectories of early training can act as replacement for exploration as rescue from off-policy states: This is never experimentally validated. This hypothesis could easiliy be validated with an ablation study, were the results of early would not be added to the replay buffer.

- High fidelity imitation: In the caption of Figure 7 the authors note that the unconditional task policy is able to outperform the demonstration videos. Thus the trajectories of the unconditional task policy allow a higher reward then the demonstrations.
Could the authors please comment on how the method still achieves high fidelity imitation even when the results of the unconditional task policy are added to the replay buffer? In prinicipal these trajectories allow a higher reward than the demonstration trajectories that should be imitated.

Mainly due to the missing experimental validation of the claims made I recommend to reject the paper.

---

> ### Author Response · Authors · 2018-11-24
> **Authors' response to reviewer 2**
>
> Thank you for taking the time to provide feedback on the paper. We have taken great care to ensure our claims are directly supported by our experiments. We will address the two claims your refer to below.
>
> > The authors claim that the method allows one-shot generalization to an unknown trajectory. To test this hypothesis the authors only provide experiments of generalization towards trajectories of a different demonstrator on the same task of stacking cubes. I would expect experiments with truly different trajectories on a different task than stacking cubes to test the hypothesis of one-shot imitation. Until then I see no evidence for a "one-shot" imitation capability of the proposed method.
>
> There is a question of terminology here, and we agree that we need to address this more precisely in the paper. Most papers on one shot imitation sample a task t~p(t) and conditional on this sample a demonstration d~p(d|t). In this setting, accomplishing t is what matters and significant deviations in the demonstration d are tolerated. In our work, we are sampling d~p(d), and for us it is important to minimize deviations in d (i.e. we want high-fidelity).
>
> Other one-shot imitation learning methods mostly focus on object diversity, e.g. pushing unseen objects or placing unseen objects (see eg the excellent website of Yu and Finn: https://bair.berkeley.edu/blog/2018/06/28/daml/ )  We instead chose a difficult control task, block stacking, which allows for many diverse ways of solving the task. We focused on demonstration diversity, following a distinctly different trajectory to solve a new task instance. While generalizing to different colours and objects is difficult, generalizing to different motions in one shot imitation is equally hard. As pointed out by Yu, Finn et al (2018) “While our work enables one-shot learning for manipulating new objects from one video of a human, our current experiments do not yet demonstrate the ability to learn entirely new motions in one shot”. Admittedly, all the different sources of variation are important and we need to make progress in all of them. Our work clearly does not address object variety, and we do need to do this in the future.
>
> > That storing the trajectories of early training can act as replacement for exploration as rescue from off-policy states: This is never experimentally validated. This hypothesis could easiliy be validated with an ablation study, were the results of early would not be added to the replay buffer.
>
> Actually, this is experimentally validated in Figure 8. Using D4PG to train the task policy, is equivalent to using our method without adding experiences from the imitation policy to the replay memory.
>
> We believe this is strong evidence our method overcomes the exploration problem, because the same RL method is used, with the same hyper-parameters, same number of actors etc, but now the transitions in the replay are more likely to see task reward.
>
> We will add a note to the paper to ensure this is clear.
>
> > High fidelity imitation: In the caption of Figure 7 the authors note that the unconditional task policy is able to outperform the demonstration videos. Thus the trajectories of the unconditional task policy allow a higher reward then the demonstrations.
> Could the authors please comment on how the method still achieves high fidelity imitation even when the results of the unconditional task policy are added to the replay buffer? In prinicipal these trajectories allow a higher reward than the demonstration trajectories that should be imitated.
>
> The task policy is able to achieve higher task reward, but at the moment we don't calculate or store its imitation reward. This is illustrated in Figure 1. The imitation policy is trying to maximize imitation reward, as a result the task policy trajectories do not interfere with the imitation policy.
>
> > Mainly due to the missing experimental validation of the claims made I recommend to reject the paper.
>
> We hope we have made clear how our claims are supported by our experiments, and that you would reconsider your evaluation. Regardless, thank you for the feedback. It has been very valuable.

---

### Official Review · AnonReviewer4 · 2018-11-13
**Interesting idea to extend DDPGfD to use only state trajectories, but needs a further experimental validation.**

**Rating:** 4
**Confidence:** 4

**Review:**

**Summary**

The paper looks at the problem of one-shot imitation with high accuracy of imitation. The main contributions:
1. learning technique for high fidelity one-shot imitation at test time.
2. Policies to improve the expert performance through RL.

The main improvements of this method is that demo action and rewards are not needed only state trajectories are sufficient.


** Comments **
- The novelty of algorithm block
The main method is very similar to D4PG-fd. The off-policy method samples from a replay buffer which comprises of both the demos and the agent experience from the previous learner iterates.

1. From a technical perspective, what is the advantage of training an imitation learner from a memory buffer of the total experience?
If the task reward is not accessed, then when the imitation learner is training, then the data should not be used for training the task policy learner. On the other hand if task reward is indeed available then what is the advantage of not using it.

2. A comparison with a BC policy to generate more experience data for the task policy agent/learning might also be useful.

* Improved Comparisons
- Compare with One-Shot Performance
Since this is one of the main contributions, explicit comparison with other one-shot imitation papers needs to be quantified with a clearly defined metric for generalization.

This comparison should be both for short-term tasks such as block pick and place (Finn et al, Pathak et al, Sermanet et al.) and also for long-term tasks as shown in (Duan et al. 2017 and also in Neural Task Programming/Neural Task Graph line of work from 2018)

- Compare High-Fidelity Performance
It is used as a differentiator of this method but without experimental evidence.
The results showing imitation reward are insufficient. The metric should be independent of the method. An evaluation might compare trajectory tracking error: for objects, end-effector, and joint positions. This is available as privileged information since the setup is in a simulation.

Furthermore, a comparison with a model-based trajectory tracking with a learned or fitted model of dynamics is also very useful.

- Compare Policy Learning Performance
In addition to D4PG variants, performance comparison with GAIL will ascertain that unconditional imitation is better than SoTA.


* Tracking a reference (from either sim or demos) is a good idea that has been explored in sim2real literature[2,3] and imitation learning [4]. It is not by itself novel. The authors fail to acknowledge any work in this line as well as provide insight why is this good and when is this valid. For instance, with highly stochastic dynamics this may not work!


- "Diverse Novel Skills"
The experiments are limited to a rather singular pick and place task with a 3-step structured reward model. It is unfair to characterize this domain as very diverse or complex from a robotics perspective. More experiments on continuous control would help.

- Bigger networks
"In fig. 3 we demonstrate that indeed a large ResNet34-style network (He et al., 2016) clearly outperforms" -- but Fig 3 is a network architecture diagram. It is probably fig 6!

- The authors are commended for presenting a broad overview of imitation based methods in table 2

** Questions **

1.  How different if the imitation learner (trained with imitation reward) from a Behaviour Cloning Policy.

2. How is the local context considered in action generation in sec 2.1.
The authors reset the simulation environment to o_1 = d_1.
Then actions are generated with  \pi_{theta} (o_t, d_{t+1}).
a. Is the environment reset every time step?
b. If not how is the deviation of the trajectory handled over time?
c. how is the time horizon for this open loop roll out chosen.

3. How is this different for a using a tracking based MPC with the same horizon? The cost can be set the same the similarity metric between states.

4. The architecture uses a deep but simplistic model. When the major attribution of the model success is to state similarity -- especially image similarity -- why did the authors not use image comparators something like the Siamese model?

Suggestion:
The whole set of experiments are in a simulation.
The authors go above and beyond in using Mitsuba for rendering images. But the images used are Mujoco rendered default. It would nice if the authors were more forthcoming about this. All image captions should clearly state -- Simulated robot results, show images used for agent training. The Mitsuba renders are only used for images but nowhere in the algorithm. So why do this at all, and if it has to be used please do it with a disclaimer. Right now this detail is rather buried in the text.

References:
1. Neural Task Programming, Xu et al. 2018 (https://arxiv.org/abs/1710.01813)
2. Preparing for the Unknown: Learning a Universal Policy with Online System Identification (https://arxiv.org/abs/1702.02453)
3. Adapt: zero-shot adaptive policy transfer for stochastic dynamical systems (https://arxiv.org/abs/1707.04674)
4. A survey of robot learning from demonstration, Argall et al. 2009

---

> ### Author Response · Authors · 2018-11-26
> **Authors' response to reviewer 4 part 1**
>
> Thank you for detailed and extensive feedback. In addition to the contribution pointed out above, we would like to emphasize that this work demonstrates that it is possible to train massive deep neural networks (larger than any previous attempt by at least an order of magnitude) for RL. Moreover, the paper shows through ablations that such architectures are essential to achieve good generalization in one-shot imitation. Smaller networks fail to generalize. We feel the problem of generalization in one-shot learning is central to AI, and as such we believe this paper presents important results showing how to advance this research frontier.  A few years ago, we certainly did not know whether RL, with its considerable variance, would allow us to train such large policies. We also did not know whether big nets were necessary at all in control tasks, as pointed out by Emo Todorov and colleagues at the previous NIPS. This paper provides empirical evidence and answers to these important questions.
>
> > The main method is very similar to D4PG-fd. The off-policy method samples from a replay buffer which comprises of both the demos and the agent experience from the previous learner iterates.
>
> For clarification, the D4PGfd algorithm was introduced in this paper, with the previous existing work --- the DDPGfg of Vecerik et al (2017) --- missing the distributed and distributional aspects of the policy optimizer.  Additionally, the D4PGfd method requires actions, but in contrast MetaMimic does not need access to actions as you note above. Moreover, MetaMimic does two things (i) one-shot high-fidelity imitation and (ii) task policy learning. The D4PGfd method only applies to task policy learning (ii). That is, it is missing an important core feature of MetaMimic.
>
> > 1. From a technical perspective, what is the advantage of training an imitation learner from a memory buffer of the total experience?
> If the task reward is not accessed, then when the imitation learner is training, then the data should not be used for training the task policy learner. On the other hand if task reward is indeed available then what is the advantage of not using it.
>
> Excellent question. The purpose of MetaMimic is twofold. The first goal is to deploy policies that users (say someone at a factory) can easily adapt, via demonstrations, to solve new tasks. Moreover, the case studied in this paper aims to meet the need for imitating the user observation trajectory with high-fidelity (eg in high precision engineering or surgery).  That is, it is not only important to accomplish the goal, but also it is important to achieve this in a very precise and specific manner.
>
> The second purpose of MetaMimic is to act as a general task policy learner by capitalizing on demonstrations of observations and rewards. Here, the process of following the demonstrations should be understood as an auxiliary task to address the problem of exploration. The final goal is a task policy. In this sense, the closest competitor to MetaMimic is DDPG-fd, but as pointed out, MetaMimic works from observations while DDPG-fd requires additional access to actions. Interestingly, our results in Figure 8, using our proposed D4PG-fd method, show that both methods perform similarly, despite MetaMimic requiring less information.
>
> We feel much of the confusion we’ve created comes from the fact that we are proposing a method that does two things. It can be useful to do (i) only, (ii) only or both (i) and (ii). It really depends on the deployment case.
>
> > 2. A comparison with a BC policy to generate more experience data for the task policy agent/learning might also be useful.
>
> This may work quite well. However, BC requires expert actions while our method does not. We already provide a strong baseline for training the task policy with access to expert actions, D4PGfD.

---

> ### Author Response · Authors · 2018-11-26
> **Authors' response to reviewer 4 part 2**
>
> * Improved Comparisons
> - Compare with One-Shot Performance
> Since this is one of the main contributions, explicit comparison with other one-shot imitation papers needs to be quantified with a clearly defined metric for generalization.
> This comparison should be both for short-term tasks such as block pick and place (Finn et al, Pathak et al, Sermanet et al.) and also for long-term tasks as shown in (Duan et al. 2017 and also in Neural Task Programming/Neural Task Graph line of work from 2018)
>
> Our paper is about one-shot high-fidelity imitation, not one-shot imitation. It is important to emphasize the high-fidelity word. That is, we want to mimic a diverse set of motions as precisely as possible. We believe this is useful in high precision engineering or surgery where departure from a very specific desired trajectory could have dire consequences. The works that we are being asked to compare against do not target high-fidelity imitation. Moreover, it is not clear they would be able to handle the diversity of motion in the demonstrations that we consider in our experiments (see the paragraph below).
>
> Yu, Finn et al (2018) on One-Shot Imitation from Observing Humans via Domain-Adaptive Meta-Learning emphasize the difficulty of being able to generalize to new motions. In their conclusion they state “While our work enables one-shot learning for manipulating new objects from one video of a human, our current experiments do not yet demonstrate the ability to learn entirely new motions in one shot”.  The authors go on to conclude “We expect that more data and a higher-capacity model would likely help enable such extensions”. This was visionary. Our experiments prove that indeed we do need much higher capacity to generalize to new motions in one-shot imitation.
>
> We highly recommend the Berkeley website of Yu and Finn on one-shot imitation at https://bair.berkeley.edu/blog/2018/06/28/daml/  The task for the first approach (first person imitation) is effectively a reaching task.  Reaching is much simpler than stacking as it does not involve reasoning about the image and dealing with contact forces. For the second approach (third person imitation) the authors do consider something closer to our approach: pick and place. As pointed out above the generalization wrt to object variety is very impressive. To train the approach requires human and robot trajectories for the same task, with the robot trajectories labelled with actions. This is great work, but solving a different problem than the one in this paper.
>
> Duan et al. (2017) sample a task t~p(t) and conditioning on this sample a demo d~p(d|t). They use a 7-DOF Fetch robot with a simple open/closed binary gripper. Importantly, for them an observation is a list of (x,y,z) object positions relative to the gripper. Unlike them, we do not assume that we know the state of the world and our input is simply pixels (learning from pixels is known to be much harder). Knowing whether the gripper is closed or open is a reasonable assumption because agents (artificial and natural) have vestibular information and proprioception. However, knowing the state of the world (positions of objects wrt gripper) is a big simplifying assumption. Note too that Duan et al. (2017) have access to actions in the demonstration sequences. The same considerations apply to the one-shot imitation NIPS paper of Wang et al (2017): https://papers.nips.cc/paper/7116-robust-imitation-of-diverse-behaviors
>
> Neural Task Programming (NTP) is an ingenious extension of the Neural Programmer-Interpreters (NPI) of Reed and de Freitas (2016), whereby task specifications in the form of video demonstrations are added to each program core. The program core predicts a sub-segmentation of the video for subsequent subprograms to process. Eventually a program API is reached, its arguments are predicted, and it is executed. The tasks demonstrated are impressive, however there are important assumptions in that work that must be taken into consideration when comparing to ours. First the bottom level programs are standard robot APIs (move_to, grip, release.move_to, etc). Even more importantly, as stated in that paper “We train the model using rich supervision from program execution traces”. Thus while we believe NTP is an important research endeavour, the setup and goals are very different from our high-fidelity one-shot imitation work.
>
> The Neural Task Graph (NTG) approach relaxes the need for supervision in terms of hierarchy, but still requires the sequences or raw visual inputs and flat actions (APIs as in NTP) for supervised training. We again see this work as being very different from ours.

---

> ### Author Response · Authors · 2018-11-26
> **Authors' response to reviewer 4 part 3**
>
> > - Compare High-Fidelity Performance
> It is used as a differentiator of this method but without experimental evidence.
> The results showing imitation reward are insufficient. The metric should be independent of the method. An evaluation might compare trajectory tracking error: for objects, end-effector, and joint positions. This is available as privileged information since the setup is in a simulation.
>
> We agree the proposed metrics would provide additional insight, thanks for the suggestion.
>
> However, we disagree we do not provide experimental evidence that our method performs high fidelity imitation. We highlight three ways we provide experimental evidence. (i) we provide evidence by generalization, we maximize the imitation reward on a training set, and show it still achieves high imitation reward on unseen demonstrations, (ii) we provide qualitative results, showing traces that illustrate the imitation policy mimics unseen demonstrations closely, in these results one can clearly see the end-effectors, joint positions, and objects closely track the demonstration, (iii) we provide a proxy metric, the task reward. Since our imitation policy is not trained using task reward, it receives no incentive to solve the task without closely imitating the demonstration.
>
> > Furthermore, a comparison with a model-based trajectory tracking with a learned or fitted model of dynamics is also very useful.
>
> Thank you for the suggestion. We will consider it for future work.
>
> > - Compare Policy Learning Performance
> In addition to D4PG variants, performance comparison with GAIL will ascertain that unconditional imitation is better than SoTA.
>
> We agree a comparison to GAIL would be quite interesting, however a fair comparison would require a bit of innovation. Vanilla GAIL is trained using on-policy RL algorithms (where as we use D4PG, an off policy method), and does not work well from images. We have tried a vanilla implementation of GAIL on this task and found it to be a very weak baseline, and think it would be unfair to include in the paper.
>
> There have been extensions proposed to GAIL to fix these limitations, including several submitted to ICLR this year, but we consider it to be beyond the scope of this paper to coalesce them into one strong baseline.
>
> > * Tracking a reference (from either sim or demos) is a good idea that has been explored in sim2real literature[2,3] and imitation learning [4]. It is not by itself novel. The authors fail to acknowledge any work in this line as well as provide insight why is this good and when is this valid. For instance, with highly stochastic dynamics this may not work!
>
> We have not claimed that tracking on its own is novel, and in fact have a section in the related work called "Imitation by tracking" which cites several papers that use tracking for imitation, including one of the earliest works in this area "Robot learning from demonstration" by Atkeson et al. (1997). In that section we highlight some of the reasons why imitation by tracking is useful: "Imitation by tracking has several advantages. For example it does not require access to expert actions at training time, can track long demonstrations, and is amenable to third person imitation". We will expand upon that section by including the citations you suggested.
>
> We agree it is quite important to test if this method works with stochastic dynamics. At the moment it is not clear whether it would. Our policy has a mechanism for compensating with drift (by comparison of its current observation and its goal observation), and it is trained to maximize its expected reward, which may be sufficient to let it deal with stochastic dynamics. We think this would be an interesting direction for future work.
>
> > - "Diverse Novel Skills"
> The experiments are limited to a rather singular pick and place task with a 3-step structured reward model. It is unfair to characterize this domain as very diverse or complex from a robotics perspective. More experiments on continuous control would help.
>
> We are planning to make this more specific: Diverse novel motions.
>
> > - Bigger networks
> "In fig. 3 we demonstrate that indeed a large ResNet34-style network (He et al., 2016) clearly outperforms" -- but Fig 3 is a network architecture diagram. It is probably fig 6!
>
> Apologies, we will fix that.
>
> > - The authors are commended for presenting a broad overview of imitation based methods in table 2
>
> Thank you.
>
> > 1.  How different if the imitation learner (trained with imitation reward) from a Behaviour Cloning Policy.
>
> Behavior cloning with a small number of training trajectories is known to have difficulty when it drifts from the distribution of states found in the dataset. This problem grows more pronounced when trajectories are long. In comparison RL with this objective helps the imitation policy stay close to the trajectories.

---

> ### Author Response · Authors · 2018-11-26
> **Authors' response to reviewer 4 part 4**
>
> > 2. How is the local context considered in action generation in sec 2.1.
> The authors reset the simulation environment to o_1 = d_1.
> Then actions are generated with  \pi_{theta} (o_t, d_{t+1}).
> a. Is the environment reset every time step?
>
> It is not. We start at the beginning of the episode and keep the environment running until a termination condition.
>
> > b. If not how is the deviation of the trajectory handled over time?
>
> The deviation of the trajectory is not handled, it is actually used as our training signal. Early in training the deviation quickly grows, making the similarly decrease, which means our agent receives lower reward. The policy is trained to maximize the expected returns, so as training progresses, the deviation of the trajectory is minimized. Doing this for a wide set of motions, and generalization to new test set motions, is far from trivial. This indeed is why we are very happy with our results.
>
> > c. how is the time horizon for this open loop roll out chosen.
>
> We roll out the policy for the length of the demonstration which is 500 time steps. We would argue that the controller is not open loop, but closed loop. The control variable is the observation. Our policy receives a target observation, and its current observation (the result of its previous action) and uses that to take a new control action.
>
>
> > 3. How is this different for a using a tracking based MPC with the same horizon? The cost can be set the same the similarity metric between states.
>
> We think this would be an interesting experiment to run. We will consider it for future work.
>
> > 4. The architecture uses a deep but simplistic model. When the major attribution of the model success is to state similarity -- especially image similarity -- why did the authors not use image comparators something like the Siamese model?
>
> It depends on what you mean. a) why did we not replace our agent architecture with features learned from a siamese model, b) why did we not replace our similarity reward with something learned by a siamese model. We address both.
>
> 	1.	We make a subtle distinction, we claim that the major component of the models success is the ability of the model to compare current observation and target observation in order to produce actions that achieve the target observation. For this reason we think training the whole network end-to-end for this purpose will perform better. If we pre-trained a siamese model to learn similarity based features, those features may not necessarily be well-suited for predicting actions, and may cap the maximum reward achievable.
> 	2.	We wanted to show our system performs quite well with a simple reward function. We agree that learning a siamese model for the reward should work quite well and mention related approaches in the paper.
>
> > Suggestion:
> The whole set of experiments are in a simulation.
> The authors go above and beyond in using Mitsuba for rendering images. But the images used are Mujoco rendered default. It would nice if the authors were more forthcoming about this. All image captions should clearly state -- Simulated robot results, show images used for agent training. The Mitsuba renders are only used for images but nowhere in the algorithm. So why do this at all, and if it has to be used please do it with a disclaimer. Right now this detail is rather buried in the text.
>
> We did this to improve presentation. We like your suggestion of adding this information to the image captions.

---

### Author Response · Authors · 2018-11-29
**Motivation for high-fidelity (over) imitation**

A few reviewers have asked us for more motivation for high-fidelity imitation. As pointed out in the paper, we were inspired by the phenomenon of over imitation in developmental psychology. In addition to the highly cited references in our manuscript,  the youtube video at (https://www.youtube.com/watch?time_continue=3&v=20Smx_nD9cw) based on Lyons, et. al. (2007) (http://www.pnas.org/content/pnas/104/50/19751.full.pdf) clearly illustrates this. In short, children, unlike other apes, choose to mimic all actions when copying an action sequence, even when those actions are irrelevant to achieving the goal. Other apes aim for the goal instead. Many developmental psychologists argue that over the years, over imitation in humans, leads to their ability to master more complex tasks. MetaMimic is a simple demonstration of this.

---

### Meta-Review · Area_Chair1 · 2018-12-16
**a novel approach for a novel task, not sufficiently grounded in prior work**

**Confidence:** 4
**Recommendation:** Reject

**Metareview:**

The paper introduces a setting called high-fidelity imitation where the goal one-shot generalization to new trajectories in a given environment. The authors contrast this with more standard one-shot imitation approaches where one-shot generalization is to a task rather than a precise trajectory. The authors propose a technique that works off of only state information, which is coupled with an RL algorithm that learns from a replay buffer that is populated by the imitator. The authors emphasize that their approach can leverage very large deep learning models, and demonstrate strong empirical performance in a (simulated) robotics setting.

A key weakness of the paper is its clarity. All reviewers were unclear about the precise setting as well as relation to prior work in one-shot imitation learning. As a result, there were substantial challenges in assessing the technical contribution of the paper. There were many requests for clarification, including for the motivation, difference between the present setting and those addressed in previous work, algorithmic details, and experiment details.

I believe that a further concern was the lack of a wide range of baselines. The authors construct several baselines that are relevant in the given setting, but did not consider "naive baseline" approaches proposed by the reviewers. For example, behavior cloning is mentioned as a potential baseline several times. The authors argue that this is not applicable as it would require expert actions. Instead of considering it a baseline, BC could be used as an "oracle" - performance that could be achieved if demonstration actions were known. As long as the access to additional information is clearly marked, such a comparison with a privileged oracle can be properly placed by the reader. Without including such commonly known reference approaches, it is very challenging to assess the proposed method's performance in the context of the difficulty of the task. Generally, whenever a paper introduces both a new task and a new approach, a lot of care needs to be taken to build up insights into whether the task appropriately reflects the domain / challenge the paper claims to address, how challenging the task is in comparison to those addressed in prior work, and to place the performance of the novel proposed method in the context of prior work. In the present paper, on top of the task and approach being novel, the pure RL baseline D4PG is not yet widely known in the community and it's performance relative to common approaches is not well understood. Including commonly known RL approaches would help put all these results in context.

The authors took great care to respond to the reviewer comments, providing thorough discussion of related work and clarifications of the task and approach, and these were very helpful to the AC to understand the paper. The AC believes that the paper has excellent potential. At the same time, a much more thorough empirical evaluation is needed to demonstrate the value of the proposed approach in this novel setting, as well as to provide additional conceptual insights into why and in what kinds of settings the algorithm performance well, or where its limitations are.